# A Critical Review of Risk Assessment Models for *Listeria monocytogenes* in Dairy Products

**DOI:** 10.3390/foods12244436

**Published:** 2023-12-11

**Authors:** Ursula Gonzales-Barron, Vasco Cadavez, Laurent Guillier, Moez Sanaa

**Affiliations:** 1Centro de Investigação de Montanha (CIMO), Instituto Politécnico de Bragança, Campus de Santa Apolónia, 5300-253 Bragança, Portugal; vcadavez@ipb.pt; 2Laboratório Para a Sustentabilidade e Tecnologia em Regiões de Montanha, Instituto Politécnico de Bragança, Campus de Santa Apolónia, 5300-253 Bragança, Portugal; 3Risk Assessment Department, French Agency for Food, Environmental and Occupational Health & Safety (Anses), 14 Rue Pierre et Marie Curie Maisons-Alfort, 94701 Maisons-Alfort, France; 4Nutrition and Food Safety Department, World Health Organization (WHO), CH-1211 Geneva, Switzerland

**Keywords:** systematic review, exposure assessment, simulation, raw milk, cheese, listeriosis

## Abstract

A review of the published quantitative risk assessment (QRA) models of *L. monocytogenes* in dairy products was undertaken in order to identify and appraise the relative effectiveness of control measures and intervention strategies implemented at primary production, processing, retail, and consumer practices. A systematic literature search retrieved 18 QRA models, most of them (9) investigated raw and pasteurized milk cheeses, with the majority covering long supply chains (4 farm-to-table and 3 processing-to-table scopes). On-farm contamination sources, either from shedding animals or from the broad environment, have been demonstrated by different QRA models to impact the risk of listeriosis, in particular for raw milk cheeses. Through scenarios and sensitivity analysis, QRA models demonstrated the importance of the modeled growth rate and lag phase duration and showed that the risk contribution of consumers’ practices is greater than in retail conditions. Storage temperature was proven to be more determinant of the final risk than storage time. Despite the pathogen’s known ability to reside in damp spots or niches, re-contamination and/or cross-contamination were modeled in only two QRA studies. Future QRA models in dairy products should entail the full farm-to-table scope, should represent cross-contamination and the use of novel technologies, and should estimate *L. monocytogenes* growth more accurately by means of better-informed kinetic parameters and realistic time–temperature trajectories.

## 1. Introduction

*Listeria monocytogenes* is a Gram-positive, non-spore-forming, facultatively anaerobic rod-shaped bacterium, pathogenic to both humans and animals and of great concern with regard to human foodborne illness [1]. Foodborne listeriosis is one of the severe foodborne diseases, and although it is a relatively rare disease, the high rate of mortality associated with this infection makes it a significant public health concern. In the European Union (EU), listeriosis was the fifth most commonly reported zoonosis in the year 2020, with 1876 confirmed cases in 27 EU Member States, and with the highest case fatality (13%) and hospitalization rates (97.1%) [1]. Recently, a meta-analysis investigation on case–control studies of sporadic listeriosis was conducted to summarize evidence on associations (odds ratios, OR) between risk factors and sporadic cases [2]. Meta-analysis models on the outcomes of 12 case–control studies spanning from 1988 to 2013 pointed seafood as the most likely causative agent of listeriosis (pooled OR = 6.27 for susceptible populations including perinatal/non-perinatal, immunocompromised, and the elderly), followed by cheese (pooled OR = 1.83; 95% CI: 1.27–2.64), mainly including soft cheeses.

In the past 20 years, many quantitative risk assessment (QRA) models have been developed worldwide in order to guide the decision making on food safety and risk management of *L. monocytogenes*. Models, from simple to more complex ones, attempt to assess in a structured approach the possible routes of contamination of *L. monocytogenes* at different points along the farm-to-fork chain of food products. Regardless of the scope of the QRA models (i.e., farm-to-table, processing-to-table, end-of-processing-to-table, retail-to-table, or consumption), their ultimate goals are to quantify the public health risk associated with the consumption of food products of interest and evaluate scenarios or potential risk reduction measures. The objectives of this study are as follows: (i) to carry out a critical review of the published risk assessment models of *L. monocytogenes* in milk and dairy products; (ii) to gather information on the type and relative effectiveness of the control measures and intervention strategies tested as what-if scenarios implemented at primary production, processing, retail, distribution, and consumer practices; and (iii) to identify key lessons, knowledge gaps and recommendations for improving future QRA models in dairy products. To meet the first objective, a systematic review was performed to retrieve the published listeriosis QRA models and extract information that could allow a transversal comparison of model structures, scenarios, and results. Although QRA models are science-based, they are unavoidably subjected to choices, value judgments, and assumptions, reasons as to why all of them must be documented in a systematic and transparent manner. Therefore, the QRA models presented in this critical review are those that contain a full description of assumptions and uncertainties.

## 2. Materials and Methods

A broad systematic review was carried out to recover published QRA models of listeriosis associated with the consumption of any foodstuff. They were sought through a literature search on Scopus and PubMed^®^, considering 1 January 1998 as the starting date of publication. The searches were carried out on 18 May 2022 (end date of publication), and the following search strings were applied for each of the literature engines:

**SCOPUS**: (TITLE-ABS-KEY (“risk assessment”) OR TITLE-ABS-KEY (exposure) OR TITLE-ABS-KEY (quantitative microbial) OR TITLE-ABS-KEY (risk modeling) OR TITLE-ABS-KEY (modeling) OR TITLE-ABS-KEY (simulation*) OR TITLE-ABS-KEY (second-order) OR TITLE-ABS-KEY (“second order”) OR TITLE-ABS-KEY (“risk management”)) AND (TITLE-ABS-KEY (“L. monocytogenes”) OR TITLE-ABS-KEY (“Listeria monocytogenes”) OR TITLE-ABS-KEY (listeriosis)).

**PUBMED**: ((“risk assessment” [Title/Abstract]) OR (exposure [Title/Abstract]) OR (quantitative microbial [Title/Abstract]) OR (risk modeling [Title/Abstract]) OR (modeling [Title/Abstract]) OR (simulation* [Title/Abstract]) OR (second-order [Title/Abstract]) OR (“second order” [Title/Abstract]) OR (“risk management” [Title/Abstract])) AND ((“L. monocytogenes” [Title/Abstract]) OR (“Listeria monocytogenes” [Title/Abstract]) OR (listeriosis [Title/Abstract])).

After merging into one single database, records were deduplicated. Eligibility assessment was carried out by two senior reviewers as a two-step process: first, by evaluating the title and abstract of the records, and subsequently, by examining the full text of the remaining records. Studies were regarded as eligible if (1) they presented a quantitative risk or exposure assessment model for listeriosis linked to any foodstuff, with formulae and assumptions explicitly indicated, and (2) they were written in English or Spanish language.

From each eligible model, the following information was extracted: scope of the QRA, food, country, existence of sub-models for cross-contamination, dose–response model and endpoint, modeling of strain variability, inclusion of temperature profiles and lag time, predictive microbiology models used, outcomes of what-if scenarios, outcomes of sensitivity analysis, and observations on the overall level of complexity of the model.

## 3. Results

### 3.1. Systematic Review Process

From Scopus, 1261 articles were collected, whereas 360 articles were retrieved from PubMed^®^. As shown in Figure 1, at the end of the full-text assessment, forty-four records were retained. In addition, through Google search, ten records were collected, which consisted of four theses [3,4,5,6], five reports from health agencies [7,8,9,10,11], and one published article [12]. A total of 54 records were therefore available, in which 65 models were published. Through this literature search, listeriosis QRA models were retrieved for any foodstuff and then classified into QRA for produce (11 models), seafood (10 models), composite (4 models), meat products (23 models), and dairy (18 models). As defined in the objectives of this study, this review focuses on dairy products only [7,8,9,10,11,12,13,14,15,16,17,18,19,20,21,22,23].

### 3.2. Description of the QRA Models in Dairy Products

A total of 18 QRA models, published from 1998, investigating dairy products as sources of listeriosis were recovered (Table 1). Out of them, nine models represented the food production conditions of Europe, covering France (Bemrah et al. [13]; Sanaa et al. [16]; Tenenhaus-Aziza et al. [12]), Italy (Giacometti et al. [20]; Condoleo et al. [15]), Greece (Koutsoumanis et al. [18]), Ireland (Tiwari et al. [14]), Denmark (Njage et al. [23]), and the EU (Pérez-Rodríguez et al. [10]). From the American continent (five QRA models), three of them pertained to the USA and Canada (FDA-FSIS [7]; Latorre et al. [19]; FDA-HealthCanada [9]), followed by Mexico (Soto-Beltrán et al. [22]) and Brazil (Campagnollo et al. [17]). Retrieved models also originated from Korea (Yang and Yoon [21]), whereas two from FAO/WHO [8] and one from EFSA BIOHAZ [11] were not particularly linked to any geographical location. Most QRA models addressed the risk of listeriosis associated with ready-to-eat (RTE) foods; only 2 out of the 18 models pertained to non-RTE food, specifically raw milk. A total of 10 out of the 18 models assessed the risk from different kinds of cheeses, and in most of the cases, they covered long supply chains; this is, either the whole supply chain (4 farm-to-table scope models) or the processing-to-table chain (3 QRA models). Six out of the nine QRA models focused on raw milk types, which, except for *queso fresco*, comprised simulations over long supply chains. This signifies that these long supply chains have been the preferred choice of the scope of QRA in cheeses as a consequence of the general understanding that *L. monocytogenes* contamination in cheeses may occur at multiple points along the food chain. The sole consumer’s practices module was simulated in two models, whereas nine models comprised end-of-processing or retail-to-table. Apart from cheese and milk, QRA models also investigated ice cream [8], yogurt [21], and cultured milk [23]. FDA-FSIS [7] investigated several RTE dairy products.

In the risk or exposure estimation procedure, most models did not perform separation of uncertainty, although the distinction was often made between uncertainty and variability distributions. Second-order simulations were undertaken in three models (16.6%). Despite the many opportunities for cross-contamination, only three QRA models attempted to characterize it. Tenenhaus et al. [12] proposed more complex discrete event models for use during cheesemaking, ripening, and packaging.

Predictive microbiology models were employed in 15 out of 18 QRA models, covering different functions for growth and survival, although the simple log-linear model for growth was often employed. Despite its importance in obtaining more realistic exposure assessment or less conservative risk estimates, not all QRA models represented the lag phase duration. The lag phase was approached in only four models (22.2% of those taking advantage of predictive microbiology). Similarly, only three QRA models solved the growth of the pathogen for time–temperature trajectories [10,12,18].

Only 1 QRA model considered mortality as an endpoint for risk estimation, whereas 18 models considered illness as an endpoint. One model did not perform any risk estimation as it only targeted the exposure assessment component. The exponential “single-hit” dose–response equation was the most frequent choice for risk characterization, being utilized in 14 QRA models (77.8%), from which the preferred approach was that of FAO/WHO [8] (9 models). The Weibull-gamma dose–response model was employed in three QRA models.

Table 2 summarizes the predictive microbiology models and main outcomes related to what-if scenarios and sensitivity analysis from the listeriosis QRA models linked to dairy products. Most of the dairy QRA models undertook separate simulations evaluating the impact of what-if scenarios, which constituted either risk factors or intervention strategies (82.4% of the models), whereas sensitivity analysis on *L. monocytogenes* concentration, on dose per serving or on risk measures as response variables was undertaken in 35.3% of the models. 

## 4. Discussion

### 4.1. Risk Factors and Control Measures Assessed at Primary Production

Many sources of contamination exist in the farm environment, such as silage, soil, water, and inadequate sanitation and housing conditions, which prompt dissemination to and between animals. In addition, *L. monocytogenes* mastitis is an important source of contamination—that increases the risk associated with contamination of raw milk—which, although it has an extremely low between- and within-herd prevalence, when present, animals may have prolonged shedding of the bacteria in the milk. *L. monocytogenes* is transmitted from animal to animal through fecal–oral routes, usually via manure contamination of the pasture or silage with the microorganism. Next to this, bulk tank milk, milk filters, milking machines, milk handlers, and poor on-farm hygiene during milking are also considered sources of contamination. The QRA model of Bemrah et al. [13], reflecting French on farm conditions at the time, showed that the contamination load due to the environment was much stronger than that of animal mastitis in the presence of *L. monocytogenes* in raw milk soft cheese. In a scenario representing lower environmental contamination that reduced the mean prevalence of contaminated farms from 3% to 2%, the median concentration of *L. monocytogenes* in raw milk cheese decreased by 99% (from 2.53 to 0.024 CFU/g) (Table 2). However, in another scenario that assumed the absence of mastitis, the median concentration of *L. monocytogenes* in raw milk cheese was reduced by only 26%—from 2.54 CFU/g (baseline scenario assuming the probability of herds with *L. monocytogenes* to be 10%) to 1.87 CFU/g.

Comparable results concerning the relative importance of mastitis, yet in sheep, were obtained from a QRA model from Italy [15], whose scenario simulations showed that the median concentration of *L. monocytogenes* in bulk tank raw milk from mastitis-free flocks decreased in only 24% (from 0.56 CFU/mL to 0.43 CFU/mL) when compared to the baseline scenario of contaminated random flocks (Table 2).

In the listeriosis QRA models available for cheese, no sensitivity analysis comparing the contributions of the environment contamination and the mastitis animals has been conducted. Only the study of Tiwari et al. [14] estimated coefficients of correlation of 0.27 and 0.15 between fecal/silage/farm contamination factors with the *L. monocytogenes* counts in raw and pasteurized milk, respectively (Table 2). Nevertheless, despite their relative importance, on-farm contamination sources, either from shedding animals or from the broad environment, have been demonstrated by different QRA models to impact the exposure dose and the risk of listeriosis, in particular from raw milk cheeses. For instance, Condoleo et al. [15] estimated that sheep’s raw milk cheeses from mastitis-free flocks presented 0.07 times the risk per contaminated serving of those from contaminated random flocks, whereas raw milk cheeses from family flocks consisting of a maximum of 10 animals each could present 8 times higher risk per contaminated serving. Similarly, increasing the initial *L. monocytogenes* population in raw milk at the farm level (between 0.03 and 10 CFU/mL for Ireland conditions) up to a maximum of 100 CFU/mL (worst-case scenario of contamination) would increase the final mean concentration of the pathogen by 35% for raw milk cheese and by 45% for pasteurized milk cheese [14]. Latorre et al. [19] tested a scenario whereby a four-fold increase in the risk per serving would occur if the prevalence of *L. monocytogenes* in bulk tank milk increased from 6% (baseline) to 25%. In the same line, according to the QRA model of FDA-Health Canada [9], a 3 log/mL reduction in *L. monocytogenes* concentration in raw milk at the beginning of cheese manufacturing—which can be interpreted as the result of the application of animal husbandry strategies for mitigating the contamination of bulk milk as raw material for cheese-making—can reduce the mean risk per serving by a factor of 7–10.

One such on-farm strategy to control the risk of listeriosis associated with raw milk is the bulk tank and tank truck milk testing in order to reduce the concentration of *L. monocytogenes* in dairy silo milk. Whereas Latorre et al. [19] estimated that a five-fold decrease in the median listeriosis annual cases for raw milk consumers would occur if a raw milk testing program were put in place (i.e., conducting monthly testing of one sample of milk and recall of milk), FDA-Health Canada [9] estimated that in raw milk soft-ripened cheeses, the milk collection testing would reduce the mean risk per serving by 24–37 times that of the risk when no testing at all is conducted (Table 2).

### 4.2. Risk Factors and Control Measures Assessed at Processing

It is widely known that pasteurization of milk is effective in destroying *L. monocytogenes*. The effectiveness of milk pasteurization as a key mitigation strategy to reduce the risk from the consumption of cheese was quantified by FDA-Health Canada [9] and FDA-FSIS [7]. The former estimated that consuming an artisanal raw milk soft-ripened cheese increased the mean risk per serving 157 times in comparison to consuming the pasteurized one in the general population. The latter estimated that the risk per serving of *queso fresco* is 43 times greater for the perinatal population and 36 times greater for the elderly population if cheeses were made from raw milk compared to pasteurized milk (Table 2).

Another strategy to control *L. monocytogenes* that can be applied during processing is the use of bacteriocinogenic lactic acid bacteria (LAB). Nevertheless, only one QRA model [17] investigated the effect of an anti-listerial cocktail from indigenous LAB on the risk of listeriosis from cheese. These authors estimated that the addition of 6 log CFU of such a LAB cocktail per ml of raw milk reduced the concentration of *L. monocytogenes* in raw milk semi-hard cheese ripened for 22 days from 7.7 log CFU/g (baseline scenario without added LABs) to 1.1 log CFU/g, which in turn reduced the risk by over 6 log. In the case of pasteurized milk soft cheeses, the addition of the same LAB cocktail to pasteurized milk inoculated at 1 log CFU/mL of *L. monocytogenes* decreased the risk 0.22-fold in both the general and vulnerable populations. Other current processing strategies, such as the smearing of cheeses with plant-based extracts having antimicrobial properties or the use of antimicrobial packaging, were not tested as what-if scenarios in any of the QRA models collected (Table 2).

### 4.3. Cross-Contamination during Processing

Despite the effectiveness of pasteurization in inactivating *L. monocytogenes*, post-pasteurization contamination and cross-contamination can occur within the processing plants and are exacerbated by the pathogen’s capacity to grow at normal refrigeration temperatures, and its ability to find damp spots or niches where they can reside and proliferate. Furthermore, if mechanical cleaning, disinfection, and rinsing are not well executed, the bacteria can form a biofilm on surfaces in contact with food, which then becomes difficult to remove by standard sanitation protocols [27]. Nonetheless, despite the relevance of cross-contamination, only two QRA models for dairy products comprised cross-contamination modules: Tenenhaus-Aziza et al. [12] and Tiwari et al. [14]. The study of Tenenhaus-Aziza et al. (12] conducted on pasteurized milk soft cheese produced in France proposes new methods for modeling cross-contamination and recontamination events. They utilize six contamination event modules, listed as follows: (1) the primo-contamination event at the cheese-making phase, whereby milk or products can be contaminated, for example, by cells from the environment or by cells arising from pasteurization failure; (2) the primo-contamination event at the ripening phase, whereby the environment of the ripening room and the smearing machine can be initially contaminated; (3) the cross-contamination during smearing, whereby a whole colony from the surface of a cheese could be transferred at a given probability to the machine or to the immediate surroundings close to the machine, through the smearing solution and the cheese matter detached from the surface of the product; (4) cross-contamination during packaging, which was modeled using the same approach but in a simplified form, where the compartments were the cheeses and the packaging machine; (5) the transfer of colonies from the smearing room to the ripening room; whereby colonies located in the environment of the smearing room are not assumed to adhere, since they come from the smearing machine and the duration between contamination of the environment and transit of the batch is not enough long to allow adherence of the cells to the environment surfaces; and (6) the recontamination during ripening, whereby during the transit of a batch inside or outside the ripening room, colonies from the environment of the ripening room can be transferred to the surface of products present in the ripening room.

This model scientifically corroborated that frequent hygienic operation is necessary in the facilities by proving that the concentration of contaminated products correlates with the total number of cells in the ripening environment. To this respect, two of the what-if scenarios estimated that when the initial number of cells in the ripening room environment decreases from 2000 to 500 cells, the mean risk of listeriosis is divided by 3.7, whereas when the primo-contamination event occurs on the smearing machine, instead of during cheese-making, with 500 cells, the mean risk is divided by 350. In the listeriosis QRA model for Irish cheeses, Tiwari et al. [14] borrowed the cheese-smearing cross-contamination model from Tenenhaus-Aziza et al. [12] and found a low correlation between the cross-contamination from the smearing machine and the counts in raw and pasteurized milk cheeses (r = 0.05 and 0.12, respectively), which sustained the scenario that if no further contamination occurred during the retail phase, but only cross-contamination due to smearing, the *L. monocytogenes* counts would decrease by 24% in raw cheeses and by 97% in pasteurized cheeses (Table 2).

In the QRA models, there is no estimate of the contribution of cross-contamination in processing plants to the final listeriosis risk. Nonetheless, it is widely known that cross-contamination is an important factor, as suggested by the many surveys throughout the world, which have reported varying prevalence levels in the environments of dairy processing plants of up to 25.0% and, as implied by the listeriosis outbreaks due to contaminated dairy products, directly linked to cross-contamination from the processing facilities [28]. Floor drains, floors, coolers, and areas of pooled water, such as washing machine areas, are sites of frequent recovery of *L. monocytogenes*, which often contaminate food contact surfaces [29]. This reinforces the importance of including sound modules to represent cross-contamination prevalence, patterns, and events in QRA models for both raw and pasteurized milk dairy products.

### 4.4. Risk Factors and Control Measures at Retail and Home

Ten dairy QRA models (58.8%) covered the contamination of *L. monocytogenes* in shorter chains (two for end-of-processing-to-table, six for retail-to-table, and two for consumption only) of dairy products, namely, raw milk (x2), pasteurized milk (x2), ice cream, yogurt, soft and semi-soft cheeses, queso fresco cheese, cultured milk, and various dairy products (Table 1). Due to the fragmented scope of these QRA models, these models address the dairy foods entering the distribution chain or the retail establishment with a certain level of contamination with *L. monocytogenes*, which is prone to multiply with prolonged storage, even from low initial concentrations. Moreover, the dairy industry’s trend toward the production of refrigerated products with longer shelf lives further aggravates this problem. Other equally important factors for increasing the risk can take place in products permitting the growth of *L. monocytogenes* during transport and distribution, retail, and home consumption, namely retail/home refrigerator temperature fluctuations and abuse, long-term storage, cross-contamination, and inadequate handling practices at retail and at home. The high frequency and amounts of dairy foods consumption also contribute to increasing the risk of listeriosis.

Many dairy QRA models compared the importance of retail/home storage temperature and retail/home storage time, and they unanimously found that the effect of higher temperature is stronger than longer time. FDA-FSIS QRA model in 10 different RTE dairy products [7] found out that if the maximum refrigerator temperature was set at 7 °C (instead of 16 °C in the baseline), the mean number of cases of listeriosis would be reduced by 69%, whereas further limiting the refrigerator temperature at a maximum of 5 °C would reduce the number of cases by >98%. On the other hand, if the maximum storage time was reduced from 14 days (baseline) to the (unrealistic) 4 days, the annual incidence of listeriosis cases would be decreased by 43.6%. In the same line of results, the listeriosis QRA model of pasteurized milk from FAO-WHO [8] undertook what-if scenarios that proved the greater relative importance of temperature over time: when the temperature distribution was shifted so that the median increased from 3.4 °C to 6.2 °C, the mean rate of illnesses increased over 10-fold for both the healthy and susceptible population; however, when the storage time distribution was extended from a median of 5.3 days to 6.7 days, the mean rate of illnesses increased 4.5-fold and 1.2-fold for the healthy and susceptible populations, respectively.

Similarly, FDA-Health Canada [9] predicted that an increase of only 1 °C in the home refrigerator temperature increases the mean risk per contaminated serving by a factor of 1.7, whereas halving the maximum duration of the home storage from 56 days to 28 days reduces the same risk by a factor of 1.4. In a more recent listeriosis QRA model from soft and semi-soft cheeses, Pérez-Rodríguez et al. [10] also compared the impact of increasing and decreasing storage temperature. When storage temperature was increased by 3–4 °C, the number of cases/million servings increased by 530%, while a decrease in the storage temperature by 3–4 °C produced only a 4% decrease in the number of cases since the baseline temperature conditions did not allow for growth *of L. monocytogenes* in cheese. By contrast, decreasing the time to consumption (storage time) by 25% produced a decrease of 33% in the incidence of listeriosis cases per million servings.

An interesting scenario was tested by Koutsoumanis et al. [18], which consisted of storing pasteurized milk cartons away from the door shelf of the fridge. According to their simulation, the proportion of cartons with no growth of *L. monocytogenes* increased from 55% to 62%. Outputs of sensitivity analysis of dairy QRA models also coincided with the relative importance of storage temperature versus storage time. Spearman rank sensitivity analysis on the probability of illness from the consumption of raw milk presented a higher correlation with the temperature of the home refrigerator (r = 0.55–0.77) than with the storage time in the home refrigerator (r = 0.27–0.36) [19]. Similarly, Tiwari et al. [14] found higher correlations of the counts of *L. monocytogenes* in raw and pasteurized milk cheeses with temperature at retail (r = 0.65 and 0.75, respectively) than with storage time at retail (r = 0.15 and 0.20, respectively) (Table 2). Nevertheless, despite the strong contribution of temperature to the risk as well as the well-known importance of maintaining the cold chain to control risks, most of the dairy QRA models (3/18) utilized variability distributions of average temperatures in the supply chain. To enable a better assessment of growth, time–temperature profiles with credible trajectories and oscillations should be used. Only three QRA models [10,12,18] solved *L. monocytogenes* growth for dynamic temperature profiles at every iteration (Table 1).

None of the retail-to-table QRA models included cross-contamination or poor handling modules, in spite of the potential of cross-contamination occurring during retail and at home. Only Pérez-Rodríguez et al. [10], when comparing the risk of listeriosis from non-sliced and sliced soft/semi-soft cheeses, indirectly determined that the processing step of slicing doubled the risk of infection, suggesting, therefore, that cross-contamination happens during slicing.

### 4.5. Contributions of Retail and Consumer Practices to the Final Risk of Listeriosis

Although not directly exposed, results from the QRA models, in perspective, pointed towards a higher contribution of the consumer module than the retail module to the risk of listeriosis from dairy foods. For instance, for the QRA model of Koutsoumanis et al. [18], the storage time at home (r = 0.482) had a stronger effect on the counts of *L. monocytogenes* in pasteurized milk at consumption than both the retail storage temperature (r = 0.181) and the retail storage time (0.174). Latorre et al. [19] also showed that the temperature of the home refrigerator (r = 0.55–0.77) can have a stronger effect than the temperature of the retail/farm refrigerator (r = 0.55) on the probability of illness from raw milk. In the case of soft-ripened cheeses [10], the risk per serving was more heavily driven by the *L. monocytogenes* counts in cheese after home storage (r = 0.95) than the counts after retail storage (r = 0.83), and in turn, than the counts after transport (r = 0.75). An interesting scenario performed in FAO/WHO [8] illustrated the strong contribution of the consumers’ practices to the risk of listeriosis by estimating that if all milk were consumed immediately after purchase at retail, the number of cases in both susceptible and healthy populations would decrease 1000-fold. All of the QRA models above coincide in that the consumers’ practices can be more determinant of the risk of listeriosis than the retail practices or conditions.

To a lesser extent, consumption as serving size or frequency has also been investigated in the dairy QRA models. Their impact on the risk is more variable, although in general, sensitivity analysis has ranked consumption-related variables lower than risk factors such as the prevalence of the pathogen, storage temperature, and time; therefore, it is less effective. According to Latorre et al. [19], the correlation between the probability of illness with serving size for raw milk purchased directly from milk tanks and milk consumed in farms was low, ranging between 0.19 and 0.30, whereas, in Yang and Yoon [21], the amount of consumption of yogurt had no effect on the risk of illness associated to drinking and regular yogurt (r = 0.08 and 0.02, respectively). In the early model of Bemrah et al. [13], it was shown that the strategy of reducing the servings per person per year of raw milk cheeses from 50 to 20 would reduce the incidence of listeriosis cases by 60%, less effective than other strategies such as excluding mastitis source or reducing the mean prevalence of *L. monocytogenes* of contaminated farms.

### 4.6. L. monocytogenes Growth Kinetic Parameters as Drivers of the Final Risk

Finally, some of the QRA models have shown that the kinetic parameters of the pathogen have a strong impact on the estimated risk. For instance, in their model for soft-ripened cheese, FDA-Health Canada [9] found moderate correlations between risk per serving and lag phase duration (r = −0.54) and exponential growth rate at 20 °C (EGR_20_) (r = 0.45); furthermore, halving the EGR_20_ of *L. monocytogenes*, decreased the mean risk per contaminated serving by a factor of ~8, and doubling the EGR_20_ multiplied the mean risk by a factor of ~4. In Tenenhaus-Aziza et al. [12], when the generation time of *L. monocytogenes* in the environment was extended from 24 h (baseline) to 48 h, the risk of listeriosis from pasteurized milk soft cheeses was divided by ~550, whilst Pérez-Rodríguez et al. [10] showed that incorporating the lag time effect to the baseline model produced a reduction of 30% in the number of cases per million servings. These findings reinforce the importance of obtaining good estimates of the microbial kinetic parameters to model the changes in microbial concentration between the point of contamination and human exposure to the pathogens.

To avoid the assumption that the *L. monocytogenes* populations are homogeneous and that their kinetic parameters represent average population behavior, the common strategy in the QRA models was to represent strain variability in parameters such as growth rate, minimum temperature for growth, minimum pH for growth and lag phase duration from growth challenge data that utilized a cocktail of *L. monocytogenes* strains [10,11,15,16,18,21]. A different approach for modeling strain variability for increased precision exposure assessment was proposed by Njage et al. [23], consisting of using whole-genome sequencing (WGS) data to unravel the biological variability that induces the diverse response by microorganisms to the differing environmental conditions. These authors employed finite mixture models to distinguish the number of *L. monocytogenes* sub-populations for each of the stress phenotypes: acid, cold, salt, and desiccation. Based on the performance assessment of the machine learning methods, they selected the support vector machine approach for the prediction of acid stress and the random forest approach for cold, salt, and desiccation stress responses. They used WGS data from a collection of 166 *L. monocytogenes* strains from Canada and Switzerland, as well as associated data on growth phenotypes during the different stress conditions. Njage et al. [23] showed that none of the four stress response categories could be represented by a unique population, instead, maximum growth rates of *L. monocytogenes* were multimodal distributions, which implied the presence of different subgroups of strains: (1) The relative growth rate distribution for the cold stress response class was estimated as consisting of 3% of cold susceptible strains (μ_max_ = 0.76) and 97% of cold tolerant strains (μ_max_ = 1.01); (2) The relative growth rate distribution for the acid stress response class was estimated as consisting of 4% of highly susceptible strains (μ_max_ = 0.41), 44% of susceptible strains (μ_max_ = 0.85), 50% of tolerant strains (μ_max_ = 1.13) and 3% of highly tolerant strains (μ_max_ = 1.50); (3) The relative growth rate distribution for the salt stress response class was estimated as consisting of 16% of susceptible strains (μ_max_ = 0.83), 77% of tolerant strains (μ_max_ = 1.01) and 7% of highly tolerant strains (μ_max_ = 1.18); and (4) The relative growth rate distribution for the desiccation stress response class was estimated as consisting of 21% of susceptible strains (μ_max_ = 0.87), 74% of tolerant strains (μ_max_ = 1.02) and 4% of highly tolerant strains (μ_max_ = 1.26). This approach, however, demands the collection of a database of strains with available WGS and phenotypic data on microbial adaptation to various inherent food characteristics and conditions encountered during food processing and handling [30].

It is crucial to bear in mind that the outcomes of a risk assessment are context-specific and influenced by factors such as the country and population under consideration. Moreover, risk assessment is inherently linked to queries posed by a risk manager. Consequently, the presentation of assessment results should be tailored to the specific question at hand, whether it involves estimating risk at the population level to gauge the overall burden or assessing risk per portion to evaluate the impact of control measures. Furthermore, certain nuances are challenging to convey accurately. For instance, the term “cheese” encompasses a diverse array of processes and microflora, rendering the transfer of models from one country to another a complex task.

Traditionally, in risk assessment, it is assumed that the matrix’s effect influences exposure, where growth or inactivation is linked to the properties of the matrix. Conversely, it is presumed that the food matrix does not influence the virulence of strains, although several studies propose that it might [31,32]. It is also generally assumed that it does not affect the variability of virulence profiles among strains. Consequently, the distribution of values characterizing intraspecific variability in the dose–response relationship is presumed to be identical regardless of the food under consideration. Nevertheless, available data demonstrate that the distribution of *L. monocytogenes* sequence types differs among various food categories [33,34]. Employing a dose–response approach that considers the diversity of virulence profiles would enable a more accurate assessment of the role of cheese and dairy products in the risk of listeriosis. A preliminary suggestion by Fritsch et al. [35] advocates for such an application.

### 4.7. Availability of Models

Sharing risk assessment models is essential to ensure the transparency of the approach and ease of re-use. This is particularly important in scientific research, where reproducibility and open science are increasingly valued [36]. By sharing models, researchers can allow others to scrutinize their work, identify any potential biases, and apply the models to their own data. This can help improve the accuracy and reliability of risk assessments and ultimately lead to better decision-making. In addition to ensuring transparency, sharing models also facilitates re-use. This is especially beneficial for researchers who may not have the resources to develop their own models [37].

Of all the studies analyzed, four provide access to the codes or spreadsheets [10,11,16,23], and one proposes to make the models used available on request [9] (details of models sharing characteristics are available in the Appendix A of this article). For the other models, no indication is given as to the availability of the models. One study dating from 2004 refers to a site that no longer exists [7]. This latter questioned the challenge of reproducibility. Indeed, after a few years, as software evolves and resources disappear (maintenance of websites, for example), it becomes difficult to reproduce the calculations made [38,39].

## 5. Conclusions

Cheese as a source of listeriosis tended to be studied in QRA models under the full farm-to-table approach because of the many factors and forces of contamination that can occur along the chain, namely, on-farm environmental contamination sources such as silage, soil, water, and inadequate sanitation and housing conditions; extensive manipulation after milk heat treatment (if heat treated); the potential for recontamination after pasteurization and cross-contamination events during processing; the possible presence of contaminating niches in processing and retail facilities; *L. monocytogenes’* ability to grow during refrigeration storage; long shelf-life in case of ripened cheeses; and wide consumption of cheese. QRA models pointed out that storage practices at home could be more determinant of the risk of listeriosis than those of retail. Furthermore, since storage temperature has a stronger effect on the risk of illness than storage time, *L. monocytogenes* growth should be more accurately estimated by using realistic time–temperature profiles as opposed to constant temperatures. Validated microbial growth kinetic parameters, as affected by lactic acid bacteria, should be employed in QRA models, given the impact of growth rate and lag phase duration on the final risk estimates. Sound cross-contamination modules that represent frequencies and contamination patterns should be placed along the food chain of dairy food. Finally, QRA models should be able to describe the effects of new non-thermal technologies, such as thermosonication and cold plasma, and bio-intervention strategies, such as ad hoc starter cultures and bioactive packaging.

## Figures and Tables

**Figure 1 foods-12-04436-f001:**
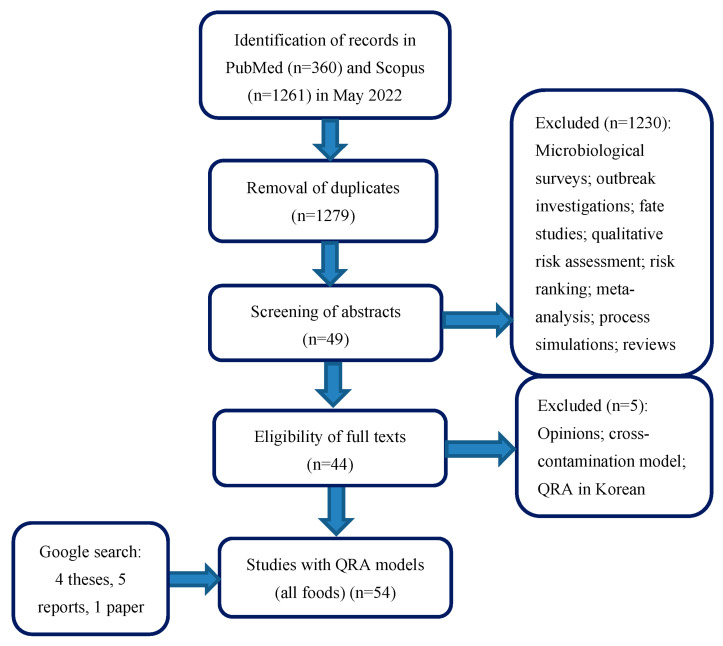
PRISMA chart of literature search of studies comprising quantitative risk assessment models on *L. monocytogenes* in foods published between January 1998 and May 2022.

**Table 1 foods-12-04436-t001:** Main features of quantitative risk assessment models of *Listeria monocytogenes* (LM) from consumption of dairy products by scope *.

Scope	Food	RTE	Cross-Contamination	DR—Endpoint	Type of DR Model	DR Sub-Populations	Strain Variability	Temp Profiles/ Lag Time	Country	Source
Farm-to-table	Raw milk soft cheese	Yes	No	WG—I, D	Farber et al. [24]	High-risk/Low-risk	Proportion of virulent strains	No/No	France	Bemrah et al. [13]
	Soft-ripened cheese	Yes	No	Exp—I	FAO/WHO [8]	High-risk/Low-risk	Strain diversity implicit in r; T_min_ and EGR_20_ represent strain variability	No/Yes	North America	FDA-HealthCanada [9]
	Raw/pasteurized milk cheese	Yes	Yes: processing, cheese smearing stage	WG—I	Bemrah et al. [13]	High-risk/Low-risk	Proportion of virulent strains	No/No	Ireland	Tiwari et al. [14]
	Sheep’s raw milk semi-soft cheese	Yes	No	Exp—I	FAO/WHO [8]	High-risk/Low-risk	Strain diversity implicit in r; challenge test data from a mixture of strains	No/No	Italy	Condoleo et al. [15]
Processing-to-table	Raw milk cheeses: Camembert of Normandy and Brie of Meau	Yes	No	Exp—I	FAO/WHO [25]	High-risk/Low-risk	T_min_ and pH_min_ represent strain variability	No/No	France	Sanaa et al. [16]
	Raw milk semi-hard cheese and pasteurized milk soft cheese	Yes	No	Exp—I	FAO/WHO [8]	High-risk/Low-risk	Strain diversity implicit in r	No/No	Brazil	Campagnollo et al. [17]
	Pasteurized milk soft cheese	Yes	Yes: cheese making (pasteurized milk, cheese surface); ripening (cross-contamination, cheese surface); packaging	Exp—I	FAO/WHO [8]	Generic	Strain diversity implicit in r; lag time distribution	Yes/Yes	France	Tenenhaus-Aziza et al. [12]
End Process-to-table	Pasteurized milk	Yes	No	.	.	.	T_min_ represents strain variability	Yes/Yes	Greece	Koutsoumanis et al. [18]
	Raw milk	No	No	Exp—I	FAO/WHO [8]	Multiple	Strain diversity implicit in r	No/No	USA	Latorre et al. [19]
Retail-to-table	Soft/semi-soft cheese	Yes	No	Exp—I	EFSA BIOHAZ [11] based on Pouillot et al. [26]	Multiple (sex/age group)	Challenge test data from a mixture of strains; strain virulence and host susceptibility explicit in r distribution	No/No	Non-specific	EFSA BIOHAZ [11]
	Various dairy products	Yes	No	Mouse Epi—I	FDA-FSIS [7]	Multiple	Variability in the virulence of different strains represented in DR	No/No	USA	FDA-FSIS [7]
	Pasteurized milk	Yes	No	Exp—I	FAO/WHO [8]	High-risk/Low-risk	Strain diversity implicit in r	No/No	Non-specific	FAO-WHO [8]
	Ice cream	Yes	No	Exp—I	FAO/WHO [8]	High-risk/Low-risk	Strain diversity implicit in r	No/No	Non-specific	FAO-WHO [8]
	Raw milk	No	No	Exp—I	FAO/WHO [8]	High-risk/Low-risk	Strain diversity implicit in r	No/No	Italy	Giacometti et al. [20]
	Soft/semi-soft cheeses	Yes	No	Exp—I	Pouillot et al. [26]	Multiple	Challenge test data from a mixture of strains; strain virulence and host susceptibility explicit in r distribution	Yes/Yes	EU	Pérez-Rodríguez et al. [10]
	Yogurt	Yes	No	Exp—I	FAO/WHO [8]	High-risk/Low-risk	Challenge test data from a mixture of strains; strain diversity implicit in r	No/No	Korea	Yang and Yoon [21]
Consumption	Raw milk cheese (Queso fresco)	Yes	No	WG—I	Farber et al. [24]	High-risk/Low-risk	Proportion of virulent strains	No/No	Mexico	Soto-Beltrán et al. [22]
	Cultured milk	Yes	No	Exp—I	FAO/WHO [25]	Multiple	Strain variability modeled by class (cold, acid, salt, desiccation stressed) from WGS data	No/No	Denmark	Njage et al. [23]

* DR: dose–response; Exp: exponential; WG: Weibull gamma; I: illness; D: death: Mouse-Epi: mouse epidemiological model; EGR_x_: exponential growth rate at x °C; r: parameter of the exponential dose–response model; T_min_: minimum temperature for microbial growth; WGS: whole-genome sequencing.

**Table 2 foods-12-04436-t002:** Predictive microbiology models and main outcomes related to what-if scenarios and sensitivity analysis from quantitative risk assessment models of *Listeria monocytogenes* (LM) from consumption of dairy products *.

Scope	Food	Predictive Microbiology Models	What-If Scenarios	Sensitivity	Complexity	Source
Farm-to-table	Raw milk soft cheese	-	(1) Excluding mastitis source of LM decreases median counts in cheese from 2.53 to 1.88 CFU/g; (2) reducing mean prevalence of contaminated farms from 3% to 2% decreases median counts in cheese from 2.53 to 0.024 CFU/g; (3) excluding mastitis source of LM and reducing the mean prevalence of contaminated farms from 3% to 2% reduces the mean incidence of listeriosis by 80%; (4) decreasing servings/person/year from 50 to 20 reduces the mean number of cases by 60%.	-	Low	Bemrah et al. [13]
	Soft-ripened cheese	Growth (linear EGR and square root, RLT); inactivation	(1) Consuming an artisanal raw milk soft-ripened cheese increases the mean risk per serving 157 times in comparison to the pasteurized one; (2) the mean risk per raw milk soft-ripened cheese serving is ~24–37 times smaller when every milk collection (bulk tank) is tested for LM, than when no testing is conducted; (3) reducing LM in raw milk by 3 log CFU/mL at the beginning of cheese manufacturing reduces the mean risk by a factor of 7–10 compared to baseline raw milk cheese; (4) testing batches of cheeses and removing non-compliant ones reduces the risk by 7–12 times that of the mean risk of non-tested pasteurized cheeses.	(1) Halving the EGR_20_ of LM reduces the mean risk per contaminated serving by a factor of ~8. Doubling the EGR_20_ multiplies the mean risk by a factor of ~4; (2) an increase of 1 °C in the home fridge temperature increases the mean risk per contaminated serving by a factor of 1.7; (3) shortening the maximum duration of the home storage from 56 to 28 days reduces the mean risk per contaminated serving by a factor of 1.4. **Outcome—risk per serving**: LM in contaminated cheese after home storage (r = 0.95); after retail storage (r = 0.83); after transport (r = 0.75); after aging (r = 0.64); LPD (r = −0.54); EGR_20_ (r = 0.45)	High: Meta-analysis, previous adjustment of mixed distributions, models for mixing and partition, lag phase modeled as “work to be done”	FDA-HealthCanada [9]
	Raw/pasteurized milk cheese	Growth (Gompertz)	(1) An increase in the initial LM in raw milk at farm level from 0.03 to 10 CFU/mL up to a maximum of 100 CFU/mL increases the final mean concentration by 35% in raw cheese and by 45% in pasteurized cheese; (2) when there is no further contamination during retail storage (only cross-contamination through smearing), the counts decrease by 24% in raw cheeses and 97% in pasteurized ones; (3) improper storage temperature above 4 °C at retail increases by 39% LM in raw cheeses and by 64% in pasteurized ones.	**Outcome—counts in raw/pasteurized milk cheese**: Temperature at retail (r = 0.65/0.75); cheese consumption (r = 0.28/0.48); storage time at retail (r = 0.15/0.20); fecal/silage/farm contamination factors (r = 0.15/0.27); cross-contamination from smearing machine (r = 0.05/0.12)	Medium: a separate Bayesian model to estimate sources of contamination on farms; discrete differential equation modeled transmission during smearing from (1) contaminated cheese to machine, (2) machine to cheese, and (3) machine to the surrounding environment and environment impact on the cheese	Tiwari et al. [14]
	Sheep’s raw milk semi-soft cheese	Growth (linear and EGR square root)	(1) Cheese from mastitis-free flocks decreased concentration of LM in bulk tank milk by 24%, in comparison to contaminated random flocks; (2) flocks with a single mastitis case increase risk per contaminated serving seven times that of contaminated random flocks; (3) cheeses from mastitis free flocks present 0.07 times the risk per contaminated serving; (4) cheeses from family flocks (10 animals maximum) have a risk eight times higher.	-	Low	Condoleo et al. [15]
Processing-to-table	Raw milk cheeses: Camembert of Normandy and Brie of Meau	Growth/no growth, Growth (modified logistic, cardinal models for temperature and pH)	-	-	Medium: growth was computed in the core and rind of cheeses considering modeled pH profiles	Sanaa et al. [16]
	Raw milk semi-hard cheese and pasteurized milk soft cheese	Growth (empirical functions)	(1) Anti-listerial LAB reduced from 7.7 log CFU LM/g of raw milk semi-hard cheese in the baseline scenario to 1.1 log CFU/g of cheese, reducing risk by >6 log RR; (2) addition of anti-listerial LAB to milk contaminated with LM at 1 log CFU/mL (the same concentration used in the baseline) reduced the risk 4.6-fold in pasteurized milk soft cheese in the general and vulnerable population.	-	Low	Campagnollo et al. [17]
	Pasteurized milk soft cheese	Growth (logistic growth model with delay and growth rate as secondary cardinal growth model with interactions)	(1) When the initial number of cells in the ripening room environment decreases from 2000 to 500 cells, the risk is divided by 3.7; (2) when the primo-contamination event occurs on the smearing machine, instead of during cheese-making, with 500 cells, the risk is divided by 350; (3) when the generation time of LM in the environment extends from 24 (base) to 48 h, the risk of listeriosis is divided by 546.	-	High: many recontamination and cross-contamination modules	Tenenhaus-Aziza et al. [12]
End Process-to-table	Pasteurized milk	Growth (linear model, polynomial functions for growth rate and lag phase duration)	(1) Changing domestic storage temperature from 5 to 4 °C increases the proportion of milk cartoons with no growth from 55 to 59%; (2) excluding the door shelf from the fridge increases the proportion of cartoons with no growth from 55% to 62%.	**Outcome—counts in milk at consumption**: Domestic storage time (r = 0.482); domestic temperature door-shelf (r = 0.288); retail storage temperature (r = 0.181); retail storage time (r = 0.174)	Medium: temperature profiles from the Greek chill chain of pasteurized milk, including transportation to retail, retail storage, and domestic storage; the lag time at a certain temperature was calculated based on the h_0_ physiological state parameter	Koutsoumanis et al. [18]
	Raw milk	Growth (linear model, square root for growth rate)	(1) Increasing LM prevalence in bulk tank milk from 6% to 25% increases the mean risk four times; (2) a five-fold decrease in the median listeriosis cases per year is observed if a raw milk testing program was in place (i.e., conducting monthly testing of one sample of milk and recall of milk).	**Outcome—probability of illness**: temperature of the home refrigerator (r = 0.55–0.77); temperature of retail/farm fridge (r = 0.55); storage time in the home refrigerator (r = 0.27–0.36); serving size for raw milk purchased directly from milk tanks and milk consumed on farms (r = 0.19–0.30)	Low	Latorre et al. [19]
Retail-to-table	Soft and semi-soft cheese	Growth (Rosso model, EGR 5 °C)	Probability of cheeses containing > 2.0 log CFU/g is 0.022 at retail and 0.024 at consumption (USA data)	Sensitivity analysis was conducted, taking together various RTE food classes.	Low: generic model; only demands some knowledge in R to utilize it	EFSA BIOHAZ [11]
	Various dairy products	Growth (linear model, square root model for EGR)	(1) If the maximum refrigerator temperature is set at 7 °C (instead of 16 °C in the baseline), the number of cases of listeriosis is reduced by 69%, and limiting the refrigerator temperature to 5 °C further reduces the number of cases to >98%; (2) reducing the maximum storage time from the 14-day baseline to 4 days reduced the annual incidence of listeriosis cases by 43.6%; (3) in *queso fresco*, the risk per serving is 43 times greater for the perinatal population and 36 times greater for the elderly population if cheeses were made from raw milk compared to pasteurized milk.	-	Medium: ten different dairy products considered; dose–response models developed for three subpopulations	FDA-FSIS [7]
	Pasteurized milk	Growth (linear model, square root growth model)	(1) If all milk were consumed immediately after purchase at retail, the number of cases in both susceptible and healthy populations would decrease 1000-fold; (2) if temperature distribution was shifted so the median increased from 3.4 to 6.2 °C, the mean number of illnesses increased > 10-fold for both populations; (3) when the storage time distribution was shifted from a median of 5.3 days to 6.7 days, the mean rate of illnesses increased 4.5-fold and 1.2-fold for the healthy and susceptible populations.	-	Medium: dose–response models for healthy and susceptible populations developed; a hierarchical beta-binomial model for the prevalence of LM in pasteurized milk	FAO-WHO [8]
	Ice cream	No growth/no death	-	-	Medium: D–R models for healthy and susceptible populations developed; a hierarchical beta-binomial model for the prevalence of LM in ice cream	FAO-WHO [8]
	Raw milk	Growth (linear model)	-	-	Low	Giacometti et al. [20]
	Soft/semi-soft cheeses	Growth (Baranyi model with Jameson effect, EGR 5 °C)	(1) Slicing the cheese increases the risk of infection by two times; (2) increasing storage temperature by 3–4 °C produces an increase of 530% cases per million servings; (3) decreasing storage temperature did not produce a substantial variation in the incidence of listeriosis (~4%) since temperature conditions in the baseline scenario did not allow for growth of LM in RTE cheese; (4) decreasing maximum mean initial LM counts values from 5 to 3 log CFU/g produces a decrease in up to 98% the cases; (5) decreasing time to consumption by 25% produced a decrease of 33% in the incidence of listeriosis cases per million servings; (6) adding the lag time effect produced a reduction of 30% in the number of cases per million servings.	-	Medium: time–temperature dynamic profiles from retail to consumption, and microbial competition models used solved with the RK4 algorithm	Pérez-Rodríguez et al. [10]
	Drinking yogurt/regular yogurt	Survival (Weibull model, secondary polynomial model)	-	**Outcome—risk of illness from drinking and regular yogurt**: prevalence of LM (r = 0.67, 0.21), storage time at market (r = −0.31,--0.35), consumption (r = 0.08, 0.02)	Low	Yang and Yoon [21]
Consumption	Queso fresco	-	-	-	Low	Soto-Beltrán et al. [22]
	Cultured milk	Growth (linear model)	-	The increase in the proportion of tolerant LM resulted in an increased association between the estimated cases per million and an increase in the concentration of the pathogen during consumer storage. This is due to the increase in the concentration of the pathogen during storage for the scenarios involving 0%, 25%, and 75% tolerant proportion groups, which were 236 ± 139, 255 ± 150, and 293 ± 172 CFU/g, respectively, compared to 274 ± 161 for the 50% tolerant proportion	High: WGS data was used to model population heterogeneity in microbial phenotypic stress responses to integrate it into predictive models	Njage et al. [23]

* LPD: lag phase duration; RLT: relative lag time; EGRx: exponential growth rate at x °C; RR: risk reduction; r: coefficient of correlation in sensitivity analysis; CFU: colony forming units; ‘-‘: absence of information in the study.

## Data Availability

Data is contained within the article. The BibTeX file containing records used in this systematic review is available.

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
