# Peer review of "A Critical Review of Risk Assessment Models for Listeria monocytogenes in Dairy Products"

_foods, 2023, doi:10.3390/foods12244436_

Round 1

Reviewer 1 Report

Comments and Suggestions for Authors

The paper is well written and structured correctly. The topics covered are certainly current and of considerable importance. The aspects of cross contamination in dairy products are extremely relevant and require further investigation.

Furthermore, quantitative risk assessment (QRA) models should better consider the specificities of the type of farming (sheep versus cattle) and those of the type of milk. In fact, different compositions affect the ability of the analytical method to recover colons of the target bacterium. I agree on the importance of cross contamination, especially for all pasteurized milk cheeses but also for whey cheeses. The authors are advised to also evaluate the importance of cheese washing machines as the main vehicle of cross contamination, especially in dairies that market raw and pasteurized milk products that are packaged and portioned in the same environment.

Author Response

We thank the reviewer for his comment. We have strengthen the discussion of the Section of Cross contamination during processing. The discussion has been expanded at the end of Section 4.3. (lines 323-333)

Reviewer 2 Report

Comments and Suggestions for Authors

This review paper presents valuable and engaging insights. Nonetheless, there are notable gaps that require attention and further elaboration. I've outlined below some of the comments that must be addressed.

Comments:

1.      This review paper presents a systematic analysis of Risk Assessment Models for Listeria monocytogenes in Dairy Products. Hence, I don't favor the term 'Critical'."

 2.      Page 1…...Introduction…………. As highlighted in the title, Listeria monocytogenes is central to your review. It would be fine if you provided some important points elucidating its characteristics and its significant role in causing foodborne listeriosis.

 3.      Page 2, line 47... L. monocytogenes should be Listeria monocytogenes (L. monocytogenes) since it's first mention. It would be more beneficial to introduce the abbreviation for the organism at this stage rather than presenting it towards the end of the manuscript, as shown by the usage of L. monocytogenes (LM) in the legend of Table 1.

 4.      Which guideline have you employed to for your systematic review (PRISMA???).

 5.      Page 2, Line 53... The objectives of this study should be revised to match the objectives of this systematic review.

 6.      Page 4, line 147... RTE should be spelled out as Ready-to-eat (RTE) since it is being used for the first time.

 7.      Page 2, Line 80... listeria should be capitalized as Listeria.

 8.      Page 3 to 4. From line 83 to 124 results and methodology sections are intertwined. Please separate them. Transfer the results to the result section. To avoid confusion, please provide separate headings for eligibility criteria as well as extraction (selection) methods you used.

 9.      Page 2, Line 93. meat products (22 models). In your title you mentioned “Dairy Products” however in your methodology you have stated as you retrieved data for all foodstuffs. It looks contradictory for readers. Please check it.

 10.  You have presented your results without any statistical value. Have you performed any statistical analysis to show the significance among the foodstuffs, the risk factors etc.

 11.  Results Page 3, line 126. …. from 1998 to which year? Please specify.

 12.  Section 4. Discussion………………. Page 4, line 176……….184. References are missing.

 13.  For tables 1 and 2, please include the explanation of the symbol '_' in the table legend. For better consistency, it would be advisable to include both CFU, FAO and WHO in the legend.

 14.  If possible, I kindly recommend the authors to make it more palatable and presentable by incorporating meta-analysis.

 15.  It would be highly beneficial for researchers and readers to include information about constituents and future directions.

Regards,

Author Response

Itemised responses in the document enclosed.

Reviewer 3 Report

Comments and Suggestions for Authors

Thank you for having an opportunity to review the manuscript entitled "A Critical Review of Risk Assessment Models for Listeria monocytogenes in Dairy Products".

After an extensive review, I am of the opinion that the manuscript need  minor revision.

It is a well-written, needed in the food testing community, and useful summary of the current status of “data publication” from a certain perspective. The discussion was particularly well written and addressed QRA challenges in each step of the food chain.

I think there is one more issue to be discussed in this paper. Namely, the likelihood of becoming unwell from consuming a certain quantity of L. monocytogenes can be accurately understood through the disease triangle, which takes into account the food matrix, the virulence of the strain, and the sensitivity of the consumer as significant considerations. Insufficient data was available about the impact of food matrix on L. monocytogenes and I would suggest adding a paragraph in the Discussion session dealing with this issue.

Also, a short address pertaining to the lack of empirical data supporting the notion that the level of risk associated with the ingestion of a specific quantity of L. monocytogenes exhibits any discernible variation across different countries when considering equivalent populations would be favorable. Disparities in manufacturing and handling protocols across different nations can potentially influence the contamination distribution, consequently impacting the risk associated with each food serving. The assessment of the public health ramifications of a particular food item can be conducted by considering two key factors: the risk associated with consuming a single serving of said food and the annual incidence of cases per given population.

Just a minor issue. There is a discongruence in the number of QRA studies, were there 17 (Line 17 and 93) or 18 (Line 126)?

Author Response

(The authors gave the same response as above.)
